# γ-Cyclodextrin Metal-Organic Frameworks: Do Solvents Make a Difference?

**DOI:** 10.3390/molecules28196876

**Published:** 2023-09-29

**Authors:** Jia X. Oh, Brent S. Murray, Alan R. Mackie, Rammile Ettelaie, Amin Sadeghpour, Ruggero Frison

**Affiliations:** 1Food Colloids and Bioprocessing Group, School of Food Science and Nutrition, University of Leeds, Leeds LS2 9JT, UK; fs18jxo@leeds.ac.uk (J.X.O.); a.r.mackie@leeds.ac.uk (A.R.M.); e.ettelaie@leeds.ac.uk (R.E.); a.sadeghpour@leeds.ac.uk (A.S.); 2Physik-Institut, Universität Zürich, Winterthurerstrasse 190, CH-8057 Zürich, Switzerland; ruggero.frisson@protonmail.ch

**Keywords:** cyclodextrin metal-organic frameworks, synthesis, characterisation, crystallinity, encapsulation

## Abstract

Conventionally, methanol is the solvent of choice in the synthesis of gamma-cyclodextrin metal-organic frameworks (γ-CD-MOFs), but using ethanol as a replacement could allow for a more food-grade synthesis condition. Therefore, the aim of the study was to compare the γ-CD-MOFs synthesised with both methanol and ethanol. The γ-CD-MOFs were characterised by scanning electron microscopy (SEM), surface area and pore measurement, Fourier transform infrared spectroscopy (FTIR) and powder X-ray diffraction (PXRD). The encapsulation efficiency (EE) and loading capacity (LC) of the γ-CD-MOFs were also determined for curcumin, using methanol, ethanol and a mixture of the two as encapsulation solvent. It was found that γ-CD-MOFs synthesised by methanol and ethanol do not differ greatly, the most significant difference being the larger crystal size of γ-CD-MOFs crystallised from ethanol. However, the change in solvent significantly influenced the EE and LC of the crystals. The higher solubility of curcumin in ethanol reduced interactions with the γ-CD-MOFs and resulted in lowered EE and LC. This suggests that different solvents should be used to deliberately manipulate the EE and LC of target compounds for better use of γ-CD-MOFs as their encapsulating and delivery agents.

## 1. Introduction

Metal-organic frameworks (MOFs) are known as highly ordered porous crystalline materials consisting of organic ligands and metal ions. Their high tunability, porosity, surface area, and thermal stability can be dictated to suit various applications [1,2,3]. Such applications include gas storage, separation science, sensors, catalysts, and drug delivery [4,5,6,7]. However, the process of synthesising MOFs were reported to be non-recyclable and highly toxic depending on the type of synthetic components or chemical reagents used [8]. To counter these limitations, a greener and more biocompatible method was investigated with the use of non-toxic alkali and alkaline metal ions in combination with organic ligand linkers of naturally occurring building blocks of amino acids, peptides and carbohydrates [9]. With the reduction in unfavourable factors in their synthesis, the applications of MOFs in the food industry and nanomedicine should be more promising.

Naturally occurring carbohydrate-based carriers, cyclodextrins (CDs) have been regarded as excellent candidates for drug-carrier preparations. In connection with this, more sustainable, edible CD-MOFs have been fabricated from γ-CDs and potassium ions via a vapour diffusion method [10]. The γ-CD-MOFs have a three-dimensional microporous structure in which the primary (C-6) hydroxyl groups are linked to potassium ions. Following the successful synthesis of γ-CD-MOFs, α- and β-CD-MOFs were also investigated and characterised [8]. However, γ-CD-MOFs have been more extensively studied due to the inherent symmetrical structure of γ-CD that gives the encapsulated compound better stability. Hence, γ-CD-MOFs have attracted growing interest as potential delivery systems for a wide range of bioactive molecules including curcumin, folic acid, resveratrol, glycyrrhizic acid, and more [11,12,13,14,15,16,17]. 

The success of various applications of γ-CD-MOFs has been said to be dependent on their crystal size since this would impact the interaction between the crystals and other molecules and their ability to be taken up by cells when being used as a delivery vehicle [18,19]. The vapour diffusion method of preparing γ-CD-MOFs at ambient temperature requires a reaction time of 7 days. The extensive time results in high polydispersity and large sizes of the crystals obtained. Much research has thus focused on the development and optimisation of synthesis methods with shortened reaction time that produce more homogenous crystals with sizes within the micron or even nanometer range [12,14,19,20,21,22]. Such methodologies require, for example, the addition of cetyltrimethylammonium bromide (CTAB) or polyethylene glycol (PEG 20,000), and/or manipulation of ratio of reactants, concentrations, and temperatures. This further complicates the synthesis method and the range of parameters to control, whilst also introducing additional compounds that are not desirable to food. 

Curcumin is a well-known natural bioactive polyphenolic compound. It has been reported to have numerous biological and pharmacological benefits, but applications of it have been limited due to its low bioavailability, generally attributed to its low water solubility and rapid metabolic conversion to inactive forms. Nevertheless, detection methods of curcumin has been well documented and many methods have been explored attempting to improve its oral bioavailability [23,24,25,26,27]. Encapsulation of curcumin in γ-CD-MOFs has been shown to be possible although much remains to be understood about the interaction between the curcumin and the crystals and the effects of the solvents used to prepare the γ-CD-MOFs or load them with this polyphenol [11,12,13]. 

In this study, the effect of the variation of two solvents, methanol and ethanol, on the synthesis of γ-CD-MOFs and their encapsulation of curcumin was investigated. With the replacement of methanol with ethanol, the process of synthesis may be made to be more food-grade, which has not been highlighted before. It was hypothesised that this change in synthesis solvent, of differing polarity, may result in the variation of crystal polymorphs obtained via the conventional vapour diffusion method. Curcumin was then encapsulated in the γ-CD-MOFs obtained, via the impregnation method, again using different encapsulation solvents. A limited amount of data is available on ethanol-synthesised γ-CD-MOFs, but a proper comparison of γ-CD-MOFs obtained synthesised using methanol (via the vapour diffusion method) has not been carried out. In addition, the effect of a variation in encapsulation solvent has never been reported. The synthesised free γ-CD-MOFs and curcumin-loaded γ-CD-MOFs were characterised by scanning electron microscopy (SEM), surface area and pore measurement, Fourier transform infrared spectroscopy (FT-IR), and powder X-ray diffraction (PXRD). To further demonstrate that the encapsulation solvent affects the encapsulation efficiency (EE) and loading capacity (LC) of γ-CD-MOFs, the encapsulation process was performed with pure methanol, pure ethanol, and mixed ratios of methanol to ethanol of 3:1, 1:1, and 1:3.

## 2. Results

### 2.1. Characterisation

A conventional 7-day vapour diffusion method of synthesising γ-CD-MOFs using methanol has reported a yield of approximately 85% [11]. In our study, with an identical methodology of synthesis, the average percentage yield was 78.40 ± 0.79%. The use of ethanol as synthesis solvent has been adopted previously but the yield was not reported [28]. The average yield of γ-CD-MOFs synthesised using ethanol in the current study was 75.53 ± 3.53%: the average yields from both solvents were found to be not significantly different (*p* > 0.05).

The γ-CD-MOF crystals appeared to have similar, block-like morphologies for both solvents (Figure 1). Such structures were also observed by Tse et al. and Kathuria et al., as opposed to regular, clear cubic structures in various other reports [10,12,13,29,30]. Both synthesis solvents resulted in crystals with a wide range of sizes as evaluated from SEM images (Figure 1). Methanol-synthesised crystals had a size range of 35–450 μm, whereas the size of ethanol-synthesised crystals was between 60 μm and 460 μm, with the average particle size of the former 165 ± 92 μm and the latter 259 ± 96 μm. Thus, the mean sizes were significantly different (*p* < 0.05). The results also agree with the size range of 40–500 μm obtained via the vapour diffusion method reported elsewhere [10].

The surface properties (i.e., surface areas and pore diameters) of free and curcumin-loaded γ-CD-MOFs are summarised in Table 1. The measured pore sizes of approximately 18.5 Å of the free γ-CD-MOFs, regardless of synthesis solvent used, were typical of a mesoporous material. The porosity of the activated γ-CD-MOFs was also verified by measuring the N_2_ gas adsorption of the samples, as shown in Figure 2. The steep increase in the low-pressure area of the isotherm is known to be distinctive to the porosity of MOFs. The Braunauer-Emmett-Teller (BET) surface area results of methanol-synthesised crystals were slightly higher than those of ethanol-synthesised but the two were not found to be significantly different (*p* > 0.05). The large reduction in the BET surface of post-encapsulation γ-CD-MOFs to approximately 1 m^2^/g is thought to be a result of curcumin occupying the pores of the crystals. 

The possible bonding between the functional groups of curcumin, free γ-CD-MOFs, and curcumin-loaded γ-CD-MOFs were examined by Fourier transform infrared (FT-IR) spectroscopy (Figure 3). The comparison of FT-IR spectra of γ-CD, γ-CD-MOFs synthesised using methanol, and γ-CD-MOFs synthesised using ethanol is shown in Appendix A. Dominant peaks observed in the FT-IR spectrum of curcumin at 3551 cm^−1^, 1626 cm^−1^, and 1602 cm^−1^ corresponded to -OH stretching, C=O stretching, and aromatic C=C stretching, respectively. Peaks expressed by free γ-CD-MOFs at approximately 3295 cm^−1^ and 1364 cm^−1^ showed a blue shift indicating that some interaction possibly occurred between curcumin and the γ-CD-MOFs post-encapsulation. A slight blue shift of -OH stretching vibration peak at 3295 cm^−1^ to approximately 3304 cm^−1^, and the shifting -OH plane bending vibration peak from 1364 cm^−1^ to approximately 1407 cm^−1^ was observed in curcumin-loaded γ-CD-MOFs. This could be inferred as attractive interactions occurring between curcumin and γ-CD-MOFs and the significance will be discussed in the next section. 

To confirm the crystallinity of the γ-CD-MOFs synthesised, powder X-ray diffraction (PXRD) of the crystals was performed. The PXRD patterns of all MOF samples as shown in Figure 4, including curcumin-loaded ones, showed a similar set of diffraction peaks, but differ from the diffraction pattern of pure curcumin. This observation indicates that curcumin molecules were incorporated within the γ-CD-MOFs. Detailed insights into the synthesised crystal structures were obtained by comparing diffraction patterns reported elsewhere [10,12,31,32]. The reflections observed at 2.87, 4.05, 4.97 and 5.73 (nm^−1^) were identified as the characteristics 110, 200, 211 and 220 from Im3m cubic structure (Figure 4). This structure was previously reported by Smaldone et al. for γ-CD-MOFs [10]. The global fitting of PXRD patterns using the CIF file (CCDC 773708) compared well with the experimental data, thus confirming the presence of Im3m cubic crystal structure (space group I432; see Figure 5a) in all the MOF samples. It should also be noted that as the results for the free unloaded γ-CD-MOFs did not show significant differences with the varying solvent, PXRD was only carried out on curcumin-loaded γ-CD-MOFs using pure methanol or ethanol as the encapsulation solvent, not mixtures of the two—see Section 2.2 below.

While the PXRD spectrum indicated that Im3m cubic crystal structure for all the synthesised MOF samples, the broadness and intensities appeared to vary from one sample to another, indicating variable crystallinity in different samples. Applying the Scherrer equation and using the broadness of 200 reflection, the crystallite sizes of all samples were estimated, and the results are summarised in Table 1 [33]. For free unloaded samples, ethanol as the synthesis solvent seemed to promote the formation of larger γ-CD-MOFs compared to methanol; however, when curcumin was involved, larger crystals were obtained in the presence of methanol as encapsulation solvent. These nanoscale crystallite size variations by PXRD corresponded well with microscale crystal dimensions from SEM data. In addition, the global fitting of diffraction patterns resulted in the determination of the lattice parameter with high precision. While both methanol- and ethanol-synthesised γ-CD-MOFs showed similar lattice parameters of 31.04 and 31.01 (±0.01) Å respectively, the curcumin-loaded samples demonstrated a notable reduction in the lattice parameters, i.e., 30.72 and 30.88 (±0.01) Å (see Figure 5b). 

### 2.2. Encapsulation Efficiency and Loading Capacity

To further investigate the effect of solvent on the encapsulation efficiency (EE) and loading capacity (LC) of γ-CD-MOFs, pure methanol, ethanol, and varying ratios of the two (methanol to ethanol = 3:1, 1:1, and 1:3) were used for the encapsulation of curcumin into the two types of crystals synthesised. Based on the results (Figure 6), a distinct advantage of methanol as an encapsulation solvent was observed. With methanol as the major encapsulating solvent, the EE was between 80–100%, regardless of the synthesis solvent used to prepare the γ-CD-MOFs. When ethanol dominated as the encapsulation solvent, a 10% reduction in EE was initially observed at a 1:3 methanol to ethanol ratio, and further halved when pure ethanol was used. The LC of the γ-CD-MOFs behaved similarly where the presence of methanol resulted in a comparable LC of approximately 30% and halved with use of pure ethanol as encapsulation solvent. However, there was no clear trend on whether the methanol- or ethanol-synthesised γ-CD-MOFs had superior EE and LC. Thus, it could be inferred that the encapsulation solvent had a greater impact on the EE and LC than the synthesis solvent.

## 3. Discussion

### 3.1. Characterisation

The consistently cuboid shapes of γ-CD-MOFs have been attributed to the ability of γ-CDs and potassium ions to form coordinate bonds in the mixed solvent environment of water and alcohol [34]. In some cases, fine particles formed by the disintegration of γ -CD-MOFs may be caused by a neutralisation step in the procedure, using weak acids [35]. Furthermore, it has been hypothesised that vacuum treatment and solvent evacuation processes may damage or collapse the structure of the crystals [20,36]. 

Larger crystals and slower crystal growth were observed here when ethanol was used as the synthesis solvent compared to methanol. Solution conditions such as supersaturation, solvent type, and temperature are known to have significant effects on crysstallisation, particularly nucleation and crystal growth. Solvent molecules can be barriers to crystal formation since they are closely associated with prenucleation solute clusters in solution [37]. A more polar solvent of greater dielectric constant and miscibility with water promotes supersaturation conditions and interfacial interactions during γ-CD-MOFs formation, which generally induces quicker rates of nuclei formation and thereby results in more uniform crystals of reduced sizes [38]. Conversely, the use of a solvent with decreased polarity would tend to cause a decrease in growth rate. This was supported by a study conducted by Liu et al. where the use of ethanol as the synthesis solvent promoted formation of hexagonal, larger-sized crystals as compared to methanol [21]. The authors also suggested that solvents with lower boiling point facilitated faster vapour diffusion, resulting in faster nuclei formation and rapid growth of smaller crystals. 

Various BET surface area values have been reported, ranging from approximately 230 m^2^/g to 1800 m^2^/g depending, on the method by which the γ-CD-MOFs were synthesised [11,14,20,39]. The large deviation in BET surface areas has been said to be caused by incomplete solvent evacuation or collapse of the crystal structure during the evacuation process [40]. The results collected from the current study fall within the range reported and are similar to those reported by Chen et al. and Lv et al., but lower than the theoretical BET surface area of 1030 m^2^/g [10,12,35]. Theoretically, any or all available surface area should be occupied by the encapsulant to maximise the encapsulation and loading efficiency of the crystals, but complete occupancy is difficult to achieve [41]. The observed large reduction of BET surface area post-encapsulation has similarly reported elsewhere, though not as drastic as seen in the current study [11,39]. For comparison, Chen et al. reported a reduction in BET surface area from 669.63 m^2^/g to 117.25 m^2^/g and 539.05 m^2^/g to 132.03 m^2^/g at an encapsulation mass concentration of γ-CD-MOFs to curcumin at 0.5:1 and 1.5:1 respectively [12]. This may be due to the further fragmentation or collapse of the γ-CD-MOFs as a result of having undergone a second solvent evacuation and vacuum treatment to collect the curcumin-loaded γ-CD-MOFs. Some fragmentation can be seen in SEM images (Appendix A).

The FT-IR spectrum of curcumin-loaded γ-CD-MOFs suggests the encapsulation solvent did not chemically change the structure of the γ-CD-MOFs. The slight shift in transmittance of curcumin-loaded γ-CD-MOFs compared to free γ-CD-MOFs from 3295 cm^−1^ to approximately 3304 cm^−1^ may be due to an interaction between the OH groups of the γ-CD-MOFs and those of curcumin. The vibration peak at 3551 cm^−1^, attributed to the phenolic hydroxyl group of curcumin, was no longer detected in post-encapsulated samples, suggesting the possibility of a hydrogen bonding with this curcumin OH group. A more distinct transmittance shift in this region in curcumin-loaded γ-CD-MOFs has also been reported previously [11,20,42]. The next most prominent shift observed was from 1364 cm^−1^ in free γ-CD-MOFs to 1407 cm^−1^ in curcumin-loaded γ-CD-MOFs. This shift may be largely due to aromatic C=C and C=O stretching (benzene skeleton) of curcumin in a similar region. It has been said that benzene rings of curcumin may be trapped within the cavities of γ-CD-MOFs via van der Waals forces and hydrophobic interactions [12]. Although it was noted that the vibration peaks at approximately 1400 cm^−1^ attributed to the alkene CH_2_ of curcumin were no longer visible, these wavenumbers overlap with those of the free γ-CD-MOFs, so little can be concluded about any specific the interactions [43]. Other studies have suggested that the disappearance of the peaks could indicate a complete inclusion of curcumin in γ-CD-MOFs. This has led to the hypothesis of curcumin being trapped in the hydrophobic region in between γ-CD pairs and cavities of γ-CD-MOF [39,42,44]. In addition, the broadness of the peaks of free γ-CD-MOFs and curcumin-loaded γ-CD-MOFs remains constant which suggests that interaction with curcumin does not affect the structure of γ-CD-MOFs. This was similarly inferred by Moussa et al. and Wang et al. [11,45].

Based on the PXRD patterns of curcumin-loaded γ-CD-MOFs, it can be inferred that the encapsulation of curcumin into γ-CD-MOFs did not disrupt the crystalline structure of the CD-MOFs, with only a slight shift in the position of the peaks (Appendix A). This non-disruptive phenomenon of γ-CD-MOFs post encapsulation has also been reported elsewhere [11,12,46]. However, the numerous sharp diffraction peaks observed in the PXRD spectrum of curcumin alone were no longer observed post encapsulation. This indicates that curcumin was successfully encapsulated into the γ-CD-MOFs, but also that the curcumin was encapsulated in an amorphous form [13,47,48]. The slight shift in the diffraction peaks could be interpreted as a shrinkage in the lattice structure as a result of encapsulation (see Figure 5b), caused by attractive interactions and if so, probably hydrogen bonding, as suggested to some extent by the FTIR spectra. Lattice shrinkage on encapsulation was also observed by Moussa et al. [11]. Such interactions have been further investigated by other researchers through molecular docking calculations. It was found that curcumin, when added at low concentrations, may preferentially occupy the hydrophobic voids of dual γ-CD units, stabilised by hydrogen bonds, followed by the formation of nanoclusters inside the large cavities of MOFs at higher concentrations [39,49]. This mechanism has also been reported in the encapsulation of folic acid and sucralose [16,35]. In our study, encapsulation was performed at a low molar ratio of γ-CD-MOFs to curcumin of 10:1 and so, hydrogen bonding between curcumin and γ-CD pairs probably dominated and this stronger interaction resulted in shrinkage of the lattice structure. 

With the presented PXRD patterns, we could not identify the exact location of curcumin molecules within the Im3m crystal. The exact atomic locations can be obtained from pair distribution function (PDF) which can only be calculated from diffraction patterns that provide access to higher order reflections (e.g., 80 reflections) and distinctly resolved diffraction peaks. 

### 3.2. Encapsulation Efficiency and Loading Capacity

The effect of the encapsulation solvent on the EE and LC of the γ-CD-MOFs may be affected by the solubility of curcumin in methanol and ethanol. The polarity of curcumin is relatively weak due to its symmetrical structure, and it is known to be more soluble in organic solvents. It was reported that curcumin, at a constant temperature (298.15 K), has better solubility in ethanol (mole fraction 0.074) compared to methanol (mole fraction 0.050) [50]. Therefore, it is reasonable to assume that curcumin more preferably interacted with ethanol as solvent rather than the hydrophobic cavity of γ-CD-MOFs, compared to methanol. It therefore makes sense that increasing the polarity of the solvent by mixing or replacement with methanol would allow better entrapment of curcumin into the cavity of γ-CD-MOFs, as observed.

In contrast, it appears that the synthesis solvent (i.e., ethanol or methanol) used to prepare the γ-CD-MOFs had little effect on the EE and LC. It is noted that the BET surface areas and therefore potential curcumin adsorption capacities of either γ-CD-MOFs were more or less the same and that post-encapsulation both types of crystal gave the same large reduction in BET surface area to 1 m^2^/g. However, the low EE and LC values measured for encapsulation via ethanol seem to contradict this. It is possible that the cavity space was occupied by both ethanol and curcumin, since curcumin has a higher affinity for ethanol than methanol. Inclusion of ethanol in γ-CD-MOFs has also been shown to be possible by Kathuria et al. [30]. Thus, the combined effect of encapsulation of both ethanol and curcumin plus possible fragmentation of crystals may be the cause of low BET surface area also observed on loading curcumin into crystals via ethanol. 

In consequence, the results therefore highlight the possible need of improving the solvent evacuation process. Control of the pressure, time and temperature profiles would be key to keeping fragmentation of γ-CD-MOFs to a minimum. The use of equipment that with more sophisticated controls would be more beneficial than a vacuum oven with manual temperature control. Different encapsulation solvents should also be further investigated to improve EE and LC.

## 4. Materials and Methods

### 4.1. Materials

γ-cyclodextrin (>90% purity), Methanol (HPLC-grade, >99.9% purity), and curcumin (from *Curcuma longa* (turmeric), powder, >65% purity) were purchased from Sigma-Aldrich (Steinheim, Germany). Potassium hydroxide (pellets, ACS reagent, >85% purity) was purchased from Fisher Scientific (Loughborough, UK). Absolute ethanol (HPLC-grade, >99.8%) was purchased from VWR Chemicals (Paris, France). Pure deionised water (with a resistivity of not less than 18.4 MΩ cm) was used in all experiments from a Milli-Q IQ 7000 system (Merck Millipore, Darmstadt, Germany). 

### 4.2. Synthesis of γ-CD-MOFs

The γ-CD-MOFs crystals were prepared via the conventional vapour diffusion method as described by Smaldone et al. [10]. Briefly, 1.30 g of γ-CD and 0.45 g of KOH (1:8 mole equivalent ratio of γ-CD to KOH) were dissolved in 20 mL of deionized water and magnetically stirred for 6 h at 600 rpm. The mixture was then filtered through a 0.45 μm organic filter membrane and sealed in a beaker containing 50 mL of methanol to allow diffusion into the sample for 7 days. The procedure was repeated with substitution of methanol to ethanol for comparison.

An activation process was carried out to free the pores of the resulting crystals from any residual moisture entrapped during synthesis. The crystals were filtered, rinsed with the corresponding solvent used for synthesis (methanol/ethanol) to remove extra unlinked potassium ions, and air-dried at room temperature for 30 min. This was followed by soaking in methanol/ethanol for 3 days before being dried under vacuum at 30 °C overnight. The crystals were either used immediately or stored in a desiccator.

### 4.3. Encapsulation of Curcumin in γ-CD-MOFs

To study the inclusion of curcumin, an impregnation method was used. A 10:1 molar ratio of γ-CD-MOFs (0.3 g) to curcumin (0.01 g) were simultaneously dispersed and dissolved in 50 mL of methanol and stirred magnetically in the dark at 450 rpm for 3 h at room temperature. Subsequently, the mixture was centrifuged (Rotina 380R, Hettich, Tuttlingen, Germany) at 5000 rpm for 30 min. The sediment was collected and washed twice with methanol to remove the unencapsulated fraction of curcumin, whether free in solution or adsorbed to the surface of γ-CD-MOFs. Finally, the curcumin-loaded γ-CD-MOFs were obtained by vacuum drying overnight at 30 °C in the dark. The procedure was similarly repeated, replacing methanol with ethanol.

### 4.4. Characterisation

#### 4.4.1. Scanning Electron Microscopy (SEM)

Morphological characterization of γ-CD-MOFs was carried out using a SEM (Fei NanoSEM Nova 450, Thermo Fisher, Loughborough, UK) operating at 3 kV. The samples were prepared by mounting crystals on a SEM stub with adhesive copper tape and then sputter depositing a layer of iridium onto the sample’s surface using a 208 HR Sputter Coater (Cressington Scientific Instruments, Watford, UK). Considering the high polydispersity of γ-CD-MOF crystals, the sizes of γ-CD-MOFs were measured with the Image Pro Plus 6.0 software (Media Cybernetics, Rockville, MD, USA).

#### 4.4.2. Surface Area and Pore Volume Measurements

The Brunauer-Emmett-Teller (BET) surface area, Langmuir surface area, and pore diameters of the activated γ-CD-MOF crystals was determined using an Accelerated Surface Area and Porosimetry System (ASAP 2020, Micromeritics, Tewkesbury, UK) equipped with nitrogen gas. Approximately 100 mg of samples were added to the sample tubes and degassed under vacuum (10^−5^ Torr) at ambient temperature for 12 h. 

#### 4.4.3. Fourier Transform Infrared (FT-IR) Spectroscopy

The FT-IR spectra of samples (curcumin, γ-CD, γ-CD-MOFs, and curcumin-loaded γ-CD-MOFs) were obtained using a Spectrum One FT-IR Spectrometer (PerkinElmer Instruments, Waltham, MA, USA) with a Golden Gate Diamond attenuated total reflection (ATR) accessory (Specac, Orpington, UK). Samples were placed directly onto the ATR diamond crystal with a set pressure of 90 cN.m applied using a torque wrench. Each spectrum was recorded with an accumulation of 100 scans at resolution of 4 cm^−1^ across a range of 550–4000 cm^−1^. 

#### 4.4.4. Powder X-ray Diffraction (PXRD)

The crystalline diffraction patterns of curcumin, γ-CD-MOFs, and curcumin-loaded γ-CD-MOFs were detected with a Bruker D8 Advance diffractometer (Bruker, Karlsruhe, Germany) at a tube voltage of 40 kV and tube current of 40 mA using Cu-Kα radiation (k = 1.5418 Å), utilising a 1.0 mm detector slit. The scans were carried out over a 2*θ* angle range of 3–45° at 0.033° increments. 

The observed intensities in the diffraction patterns were fitted using the program FullProf using the LeBail method where a general constant scale factor was used to match the observed intensities while adjusting the zero-error, lattice parameter, and the peak broadening parameters [51]. The background was adjusted with a cubic spline interpolation of a set of user-provided points. This ensured convergence of the fit within a few cycles. The 3D crystal structure is visualised in Mercury environment using the Crystallographic Information File (CIF) deposited on the Cambridge Structural Database (CCDC; 773708) by Smaldone et al. [10]. The same file was used to fit the diffraction patterns. 

### 4.5. Encapsulation Efficiency and Loading Capacity

The amount of unencapsulated curcumin in the supernatant collected as described in Section 4.3 was calculated from a calibration curve using the SPECORD 210 Plus UV-Vis spectrophotometer (Analytik Jena, Jena, Germany) at 426 nm. Ratios of methanol to ethanol of 3:1, 1:1, and 1:3 were also investigated on their effect on encapsulation. The encapsulation efficiency (EE, %) and loading capacity (LC, %) of curcumin in inclusion complexes were subsequently determined by using the following equations [12]:(1)$$EE(\%)=\frac{Amountofencapsulatedcurcumin}{Totalamountofcurcumin}\times 100\%$$
(2)$$LC(\%)=\frac{Amountofencapsulatedcurcumin}{TotalamountofcurcuminandCD-MOFs}\times 100\%$$

### 4.6. Statistical Analysis

Data were collected in triplicate and the results were calculated and expressed as mean ± standard deviation where applicable. Statistical analysis was carried out using OriginPro 2023b (OriginLab, Northhampton, MA, USA). Statistical significance was established at *p* < 0.05. 

## 5. Conclusions

In this investigation, a comparison between the solvents, methanol and ethanol, in the synthesis of γ-CD-MOFs was made. It was found that the only significant difference the solvent made was the larger crystal size with ethanol. This is largely due to the lower polarity of this solvent compared to methanol and its effects on the rate of nuclei formation and uniformity of the crystals formed. The crystalline structures and BET determined surface areas of both sets of γ-CD-MOFs appeared to be the same. This suggested that the preferable use of ethanol as a solvent could be used to prepare γ-CD-MOFs rather than the more usual methanol. However, variation of encapsulation solvent made a significant difference in entrapping curcumin into the cavities of γ-CD-MOFs: the use of methanol promoted the encapsulation process. When ethanol dominated as the encapsulation solvent, a drastic drop in EE and LC was observed, due to the higher solubility of curcumin in ethanol which allowed for less interaction with the hydrophobic cavities of the crystals. Hence, different solvents should be further investigated to deliberately manipulate the EE and LC of target compounds for better use of γ-CD-MOFs as their encapsulating and delivery agents. 

## Figures and Tables

**Figure 1 molecules-28-06876-f001:**
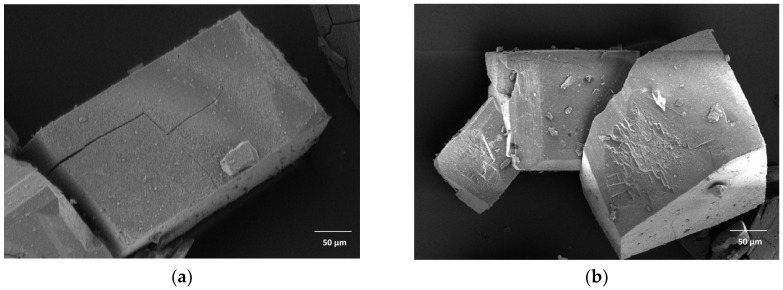
Scanning electron microscopy (SEM) morphology images of γ-cyclodextrin metal-organic frameworks (γ-CD-MOFs) prepared with different synthesis solvent: (**a**) methanol and (**b**) ethanol, respectively.

**Figure 2 molecules-28-06876-f002:**
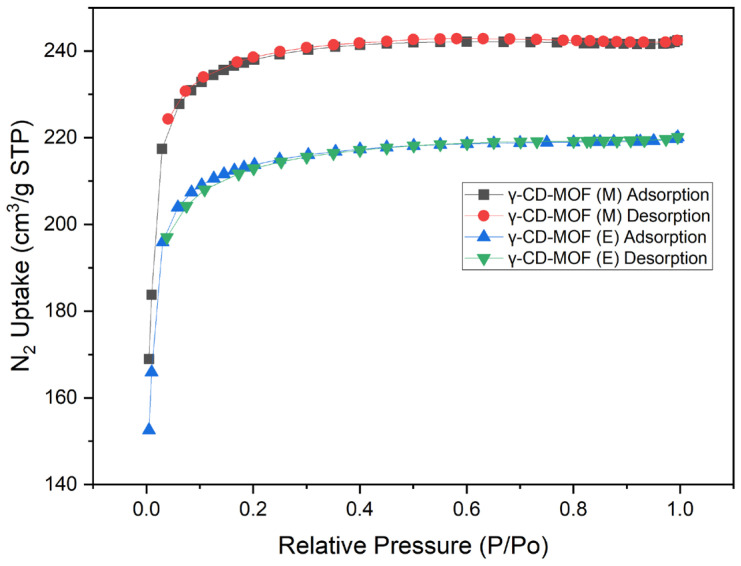
Nitrogen adsorption-desorption isotherms of activated free γ-CD-MOFs synthesised using either methanol (M) or ethanol (E).

**Figure 3 molecules-28-06876-f003:**
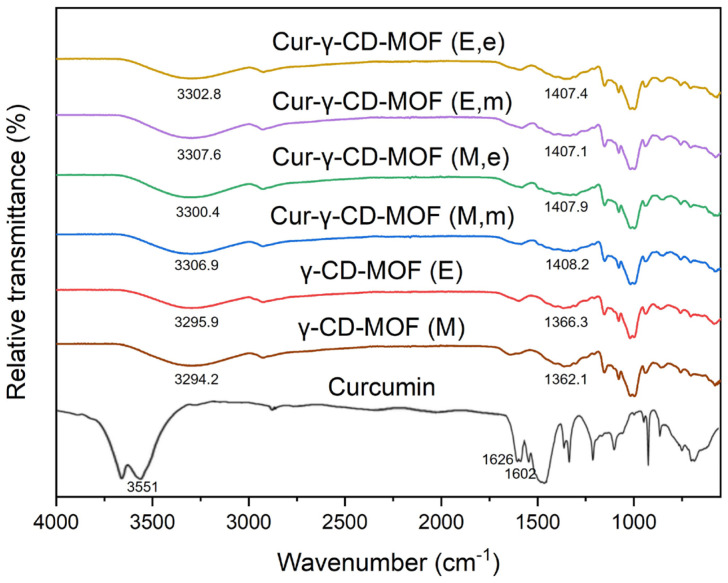
Fourier transform infrared spectroscopy (FT-IR) patterns of curcumin, free γ-CD-MOFs, and curcumin-loaded γ-CD-MOFs. (M) denotes crystals synthesised with methanol, and (E) with ethanol; (m) indicating that methanol was used as encapsulation solvent, and (e) as ethanol.

**Figure 4 molecules-28-06876-f004:**
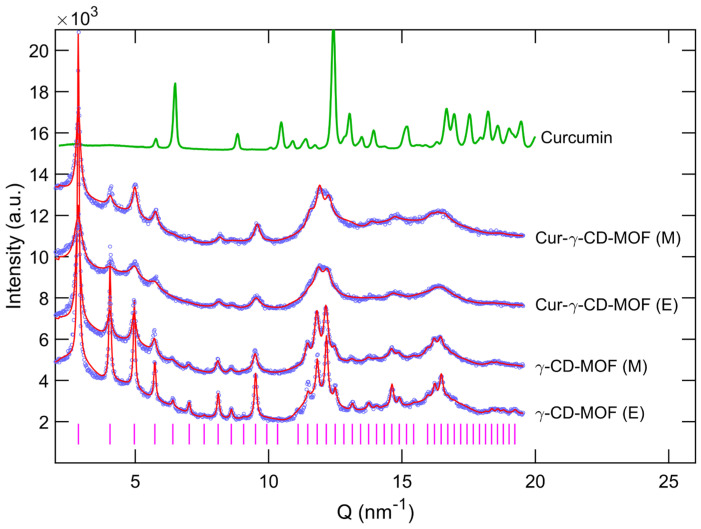
Powder X-ray diffraction patterns of γ-CD-MOFs synthesised with methanol (M) and ethanol (E) as synthesis solvent and curcumin encapsulated γ-CD-MOFs using corresponding encapsulation solvent of methanol and ethanol. The experimental diffraction patterns of all MOF samples are shown in blue while their corresponding theoretical fitted curves are shown by solid red lines. The pink dashed lines show the 64 reflections from the Im3m cubic crystal structure (space group I432). The diffraction pattern from pure curcumin powder is shown in green line. The *x*-axis represents the scattering vector magnitude equivalent to 4π/λ×sin(θ). The diffraction curves are shifted vertically for better clarity.

**Figure 5 molecules-28-06876-f005:**
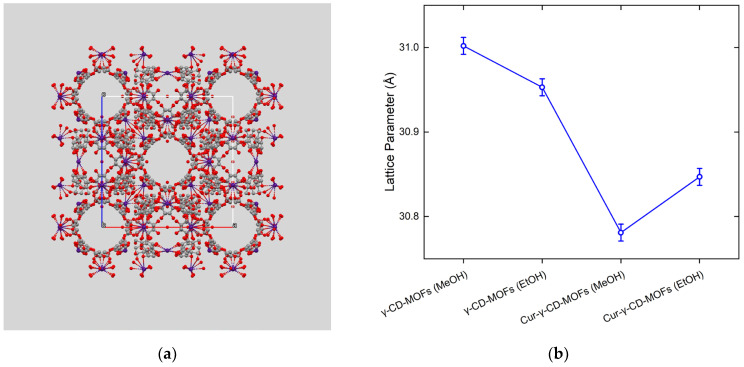
(**a**) Representation of crystal structure of γ-CD-MOFs observed in our study using methanol and ethanol as synthesis and encapsulation solvents in absence/presence of curcumin. All synthesised MOFs demonstrated an Im3m cubic symmetry (I432 space group). (**b**) Different lattice dimensions evaluated for samples synthesised and encapsulated with different solvents. Noticeable reduction in lattice parameter is observed for curcumin-loaded samples.

**Figure 6 molecules-28-06876-f006:**
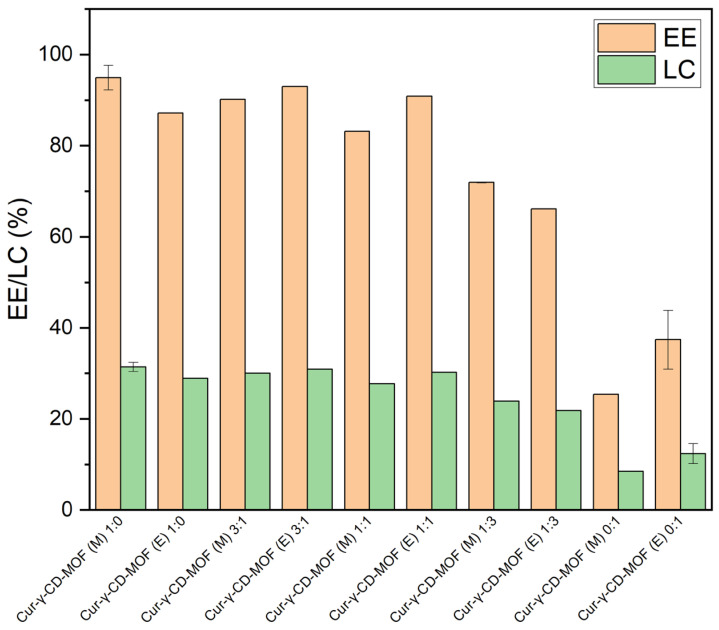
Encapsulation efficiency (EE) and loading capacity (LC) of methanol- and ethanol-synthesised γ-CD-MOFs in varying ratios of encapsulation solvents of methanol to ethanol at 1:0, 3:1, 1:1, 1:3 and 0:1.

**Table 1 molecules-28-06876-t001:** Surface parameters and crystallite sizes obtained from Scherrer equation using 200 reflection of the γ-CD-MOFs prepared with varying synthesis solvent and curcumin-loaded γ-CD-MOF (Cur- γ-CD-MOF) crystals prepared with varying encapsulation solvent.

Samples	BET Surface Area (m^2^/g)	Langmuir Surface Area (m^2^/g)	Pore Diameters (Å)	Crystallite Size (nm)
CD-MOF (M)	799.4 ± 14.6 ^a^	813.7 ± 14.3 ^a^	18.6 ± 0.1 ^a^	398 ± 38
CD-MOF (E)	719.1 ± 13.4 ^a^	848.1 ± 10.2 ^a^	18.7 ± 0.2 ^a^	750 ± 56
Cur-CD-MOF (M, m)	1.6 ± 0.4 ^b^	-	-	372 ± 76
Cur-CD-MOF (M, e)	1.3 ± 0.2 ^b^	-	-	-
Cur-CD-MOF (E, m)	1.7 ± 0.6 ^b^	-	-	-
Cur-CD-MOF (E, e)	1.6 ± 0.5 ^b^	-	-	270 ± 89

Letters in brackets: M or E indicates that the synthesis solvent used in preparation of γ-CD-MOFs was either methanol or ethanol; m or e indicates that solvent used for encapsulation was either methanol or ethanol. Values in the same column having different letters were significantly different (*p* < 0.05).

## Data Availability

Data is contained within the article or Appendix A.

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
