# Peer review of "γ-Cyclodextrin Metal-Organic Frameworks: Do Solvents Make a Difference?"

_molecules, 2023, doi:10.3390/molecules28196876_

Round 1
Reviewer 1 Report
The aim of the study is focuses on the difference of γ-CD-MOFs synthesized when using methanol or ethanol as the reaction solvent during synthesis, and the difference of using methanol and ethanol as encapsulators when packaging curcumin into γ-CD-MOFs. Compared with the traditional γ-CD-MOFs synthetic solvent methanol, ethanol is no different from methanol in the process of synthetic MOFs, and ethanol is harmless to biology and can prepare food grade MOFs material for biomedicine. This work is worthy of publication in molecules if the following comments are suitably handled during the revision.
1. The PXRD standard card of γ-CD-MOFs should be placed into the figure as a reference.
2. Thermal weight analysis can be used to explore the content of curcumin encapsullated in γ-CD-MOFs and also to verify successful encapsulation of curcumin.
3. The BET value of γ-CD-MOFs (E) is less than that of γ-CD-MOFs (M), but in Figure 6, when varying ratios of encapsulation solvents of methanol to ethanol at 3:1 and 1:1, the Encapsulation efficiency (EE) and loading capacity (LC) of γ-CD-MOFs (M) are less than γ-CD-MOFs(E), Why.?
4. Recent references to this catalytic reaction should be added.
Author Response
Comments and Suggestions for Authors
The aim of the study is focuses on the difference of γ-CD-MOFs synthesized when using methanol or ethanol as the reaction solvent during synthesis, and the difference of using methanol and ethanol as encapsulators when packaging curcumin into γ-CD-MOFs. Compared with the traditional γ-CD-MOFs synthetic solvent methanol, ethanol is no different from methanol in the process of synthetic MOFs, and ethanol is harmless to biology and can prepare food grade MOFs material for biomedicine. This work is worthy of publication in molecules if the following comments are suitably handled during the revision.
- The PXRD standard card of γ-CD-MOFsshould be placed into the figure as a reference.
As we have opted to do a global fitting using an existing CIF file from a published paper, we feel that the reported diffraction peaks within the CIF file have been sufficiently represented in our figure (Figure 4). Hence, we have decided not to overcrowd the figure with too many data lines.
We have now added the name and where the referenced crystallographic data can be accessed within the article in line 159, “… using the CIF file (CCDC 773708) compared well…” This has also been mentioned within the Materials and Methods section.
- Thermal weight analysis can be used to explore the content of curcumin encapsullated in γ-CD-MOFs and also to verify successful encapsulation of curcumin.
To the best of our knowledge, thermal gravimetric analysis (TGA) can be used to assess if curcumin has been successfully encapsulated in γ-CD-MOFs, as does the Fourier transform infrared (FTIR) spectroscopy, but neither can accurately predict the content of curcumin within the γ-CD-MOFs. Moreover, the FTIR results have provided information on how the curcumin has interacted with γ-CD-MOFs when being encapsulated. As the characterisation experiments have not proven that the crystals were significantly different with variation of synthesis solvent, we did not think that TGA would provide additional information.
- The BET value of γ-CD-MOFs (E) is less than that of γ-CD-MOFs (M), but in Figure 6, when varying ratios of encapsulation solvents of methanol to ethanol at 3:1 and 1:1, the Encapsulation efficiency (EE) and loading capacity (LC) of γ-CD-MOFs (M) are less than γ-CD-MOFs(E), Why.?
Statistical analysis (one-way anova) has shown that although the EE and LC of γ-CD-MOFs (M) were lower than γ-CD-MOFs (E), the two were not significantly different with ratio of methanol to ethanol at 3:1 and 1:1, which led us to conclude that there was no clear trend on whether which of the two had superior EE and LC.
It is possible that the slightly lowered EE and LC were at least partly due to the fragmentation of crystals as a result of solvent evacuation process during synthesis. As the γ-CD-MOFs were neither carefully selected for BET analysis nor for encapsulation processing, we were not able to confirm if the crystals were largely whole or fragmented to begin with.
- Recent references to this catalytic reaction should be added.
References regarding the encapsulation of curcumin not affecting the structure/crystallinity of γ-CD-MOFs has now been added.
(line 273) “…not affect the structure of γ-CD-MOFs. This was similarly inferred by Moussa et al and Wang et al. [10,42].”
(line 277) “…position of the peaks (Figure S3). This non-disruptive phenomenon of γ-CD-MOFs post encapsulation has also been reported elsewhere [10,11,43].”
Submission Date
22 August 2023
Date of this review
07 Sep 2023 05:31:37
Reviewer 2 Report
The article describes the influence of solvents in the synthesis of γ-Cyclodextrin Metal-Organic Frameworks.
The theme is interesting, although some concerns must be addressed:
- Some newer references must be introduced in the Introduction” section.
- Please introduce the figures after their first mention in the manuscript.
- Line 141: Please replace the verb “saw” with a more appropriate one.
- Figure 1 caption must be shortened, part of the text may be introduced as a discussion in the manuscript.
- Please revise Figure 5 (b): the fonts are way too big compared to the graph.
- Full information about the used reagents and instruments (company, city, country) must be provided.
- Please specify whether the samples were dried before the FTIR analysis; in case of a positive response, please specify the exact conditions.
Minor editing must be achieved in the English language.
Author Response
Comments and Suggestions for Authors
The article describes the influence of solvents in the synthesis of γ-Cyclodextrin Metal-Organic Frameworks.
The theme is interesting, although some concerns must be addressed:
- Some newer references must be introduced in the Introduction” section.
Newer references (published within the last 5 years) have now been added to support or replace cited references, where appropriate. The “References” section has also been updated with a reference manager.
- Please introduce the figures after their first mention in the manuscript.
A correction has been made for the first mention of Figure 4 in line 152: “The PXRD patterns of all MOF samples as shown in Figure 4, including curcumin-loaded ones, showed a similar set of diffraction peaks, but differ from the diffraction pattern of pure curcumin.”
An error on figure number within text has also been corrected for Figure 6 in line 199.
- Line 141: Please replace the verb “saw” with a more appropriate one.
The sentence has been restructured to “A slight blue shift of -OH stretching vibration peak at 3295 cm-1 to approximately 3304 cm-1, and the shifting -OH plane bending vibration peak from 1364 cm-1 to approximately 1407cm-1 was observed in curcumin-loaded γ-CD-MOFs.”
- Figure 1 caption must be shortened, part of the text may be introduced as a discussion in the manuscript.
We believe that the caption concisely describes Figure 1 and shortening it would reduce the clarity and may cause confusion, since readers would need to search within the paragraph to look for the relevant description.
- Please revise Figure 5 (b): the fonts are way too big compared to the graph.
Figure 5(b) has now been replaced with adjustments made to fonts.
- Full information about the used reagents and instruments (company, city, country) must be provided.
Additional info (city and country) has been added for reagents and instruments listed within the “Materials and Methods” section.
- Please specify whether the samples were dried before the FTIR analysis; in case of a positive response, please specify the exact conditions.
The sequence of preparation of samples are as written in the “materials and methods” section 4.2 and 4.3. γ-CD-MOFs were first synthesised and activated, followed by an overnight vacuum drying process at 30°C. After encapsulation of curcumin in the γ-CD-MOFs, the encapsulated samples were again dried overnight at 30°C. All the samples (unloaded/free or encapsulated γ-CD-MOFs) were in a dried powdered form before being used in any characterisation experiments.
Comments on the Quality of English Language
Minor editing must be achieved in the English language.
The article has already been proof read again and any minor errors corrected.
Submission Date
22 August 2023
Date of this review
08 Sep 2023 12:14:55
Reviewer 3 Report
The paper reports on the comparison of the syntheses of CD-MOFs using two solvents: conventional methanol and ethanol. A complete characterization of the samples before and after loading with curcumin was carried out.
I believe the work could be of interest for wide audience. However, some minor misprints could be corrected:
1) line 50: oxford comma missing
2) line 52: "The success of various applications of CD-MOFs has been said to be dependent on their crystal ...". Crystal structure? crystal morphology?
3) line 123 "Surface parameters and and ..."
4) line 223: "effects on cyrstallisation"
5) line 371: Approximately 100 g of sample..." Maybe 100 mg?
Author Response
Comments and Suggestions for Authors
The paper reports on the comparison of the syntheses of CD-MOFs using two solvents: conventional methanol and ethanol. A complete characterization of the samples before and after loading with curcumin was carried out.
I believe the work could be of interest for wide audience. However, some minor misprints could be corrected:
1) line 50: oxford comma missing
This has now been added “… curcumin, folic acid, resveratrol, glycyrrhizic acid, and more.”
2) line 52: "The success of various applications of CD-MOFs has been said to be dependent on their crystal ...". Crystal structure? crystal morphology?
The statement is referring to the sizes of the crystals and has now been amended “…dependent on their crystal size…”
3) line 123 "Surface parameters and and ..."
Corrected.
4) line 223: "effects on cyrstallisation"
Spelling error corrected.
5) line 371: Approximately 100 g of sample..." Maybe 100 mg?
Yes, and have now been amended.
Submission Date
22 August 2023
Date of this review
06 Sep 2023 18:03:42